# Imaging performance of portable and conventional ultrasound imaging technologies for ophthalmic applications

Jack O. Thomas[1,2,☯], Josiah K. To[1☯], Anderson N. Vu[1☯], David Horton[3], Ermin Dzihic[3], Andrew W. Browne[1,4,5]*

1 Gavin Herbert Eye Institute, Department of Ophthalmology, University of California Irvine, Irvine, California, United States of America, 2 School of Medicine, California University of Science and Medicine, Colton, California, United States of America, 3 School of Medicine, University of California Irvine, Irvine, California, United States of America, 4 Department of Biomedical Engineering, Henry Samueli School of Engineering, University of California Irvine, Irvine, California, United States of America, 5 Institute for Clinical and Translational Science, University of California Irvine, Irvine, California, United States of America

☯ These authors contributed equally to this work.
* abrowne1@hs.uci.edu

**Data Availability Statement:** All relevant data are within the manuscript and its Supporting Information files.

## Abstract

### Purpose

The aim of this study was to evaluate the imaging capabilities of Butterfly iQ with conventional ophthalmic (piezoelectric) ultrasound (COU) for ophthalmic imaging.

### Methods

Custom phantom molds were designed and imaged with Butterfly iQ and COU to compare spatial resolution capabilities. To evaluate the clinical imaging performance of Butterfly iQ and COU, a survey containing pathological conditions from human subjects, imaged with both Butterfly iQ and COU probes, was given to three retina specialists and graded on image detail, resolution, quality, and diagnostic confidence on a ten-point Likert scale. Kruskal-Wallis analysis was performed for survey responses.

### Results

Butterfly iQ and COU had comparable capabilities for imaging small axial and lateral phantom features (down to 0.1 mm) of high and low acoustic reflectivity. One of three retina specialists demonstrated a statistically significant preference for COU related to resolution, detail, and diagnostic confidence, but the remaining graders showed no significant preference for Butterfly iQ or COU across all sample images presented.

### Conclusion

The emergence of portable ultrasound probes offers an affordable alternative to COU technologies with comparable qualitative imaging resolution down to 0.1 mm. These findings suggest the value to further study the use of portable ultrasound systems and their utility in routine eye care.

**Funding:** Research to Prevent Blindness Grant #: None Recipient: University of California Irvine Department of Ophthalmology BrightFocus Foundation Award#: M2021013N Recipient: Andrew W. Browne Arnold and Mabel Beckman Foundation Award#: None Recipient: University of California Irvine Department of Ophthalmology The funders had no role in study design, data collection and analysis, decision to publish, or preparation of the manuscript.

**Competing interests:** The authors have declared that no competing interests exist.

# Introduction

Ophthalmic ultrasound is an important imaging modality for visualizing ocular anatomy. Recent advancements in ultrasound technology offer increased portability with decreased equipment costs, potentially expanding access to ultrasound imaging. Importantly, when leveraging modern-day ultrasound machines without the use of contrast agents, no known adverse events have been documented in humans [1]. Currently, ophthalmic ultrasound is used to evaluate anterior segment anatomy, intraocular anatomy when media opacity obscures optical examination, intraocular foreign bodies, solid tumors, and periocular anatomy in the orbit [2, 3]. Limited access to conventional ultrasound systems due to high costs and lack of portability poses significant challenges for providing basic ophthalmic care in low-income countries [4, 5]. Some portable ultrasound systems costing approximately $2,699 have emerged and introduce cost-effective competition to ophthalmic ultrasound systems costing $20,000–70,000 (Table 1).

To better understand the potential impact of portable and cost-effective ultrasound systems on ophthalmic care, it is important to evaluate the spatial resolution capabilities of these devices. Spatial resolution is defined as a device's ability to distinguish between two points at a particular depth and is central to an ultrasound system's ability to visualize anatomical features. Spatial resolution of ultrasound images is comprised of two components: axial and lateral resolution. Axial resolution describes the minimum distance that can be differentiated between two reflectors located parallel to the direction of the ultrasound beam propagation. Lateral resolution is the minimum distance that can be distinguished between two reflectors that are separated perpendicular to the direction of the sound beam [6].

Some emerging portable ultrasound systems use a fundamentally different type of ultrasound technology compared to conventional ultrasound instruments. Ultrasound images are created by generating ultrasonic waves and reflecting them off tissue interfaces. The elapsed time between generating and detecting reflected sound waves is used to create a 2D gray-scale image [7, 8]. In conventional ultrasound instruments, ultrasonic waves are produced by applying a current to a piezoelectric crystal [8]. In contrast, the Butterfly iQ (Butterfly Network Inc, Guilford, CT) uses a unique silicon "ultrasound-on-a-chip" that contains a 2D array of 9,000

**Table 1. Comparisons of commercially available handheld ultrasound instruments.**

| Device | Manufacturer | Transducer | Base Unit Price (USD) | Yearly Subscription | System compatibility | Approved for ophthalmic use |
|---|---|---|---|---|---|---|
| iQ | Butterfly | CMUT | $2699 | $420/year | iOS/Android | Yes |
| Vscan Extend R2 | GE | Piezoelectric | $4995 | - | Provided tablet | Yes |
| L20 | Clarius | Piezoelectric | $6900 | - | iOS/Android | Yes |
| iViz | Sonosite | Piezoelectric | Discontinued | - | Provided Tablet | L38V and L25v probe required |
| 300L | Healcerion | Piezoelectric | $4,995 | - | iOS/Android | No |
| Lumify | Philips | Piezoelectric | $6,000 or subscription | $2388/year | iOS/Android | No |
| Biim | Biim | Piezoelectric | $2000–5000 | - | iOS/Android | No |
| L38-22 & L22-8 | Kolo Medical | CMUT | $19,750 (L38-22v) $18,750 (L22-8v) | - | Requires Verasonic Vantage instrument | No |
| 4Sight | Accutome | Piezoelectric | $20,000 | - | Provided Tablet | Yes |
| ABSolu | Quantel Medical | Piezoelectric | $70,000 | - | Provided Tablet | Yes |

Prices adapted from manufacturer websites or representatives.

capacitive micromachined ultrasonic transducers (CMUTs) [9–11]. CMUT units are composed of a vacuum gap that separates a conductive plate or membrane over a substrate, thus acting like a capacitor [12]. When alternating current is applied, the unit begins to emit ultrasonic sound waves [10, 12]. Although largely explored in non-medical contexts, such as non-destructive testing in aerospace engineering, CMUTs are now being adapted for healthcare applications [12, 13].

Overall, large-scale changes in ultrasound technology have revolutionized its cost and adaptability. The incorporation of CMUT arrays has improved ultrasound resolution by maintaining a higher sensitivity and wider bandwidth for emission and detection [10–12, 14]. Additionally, CMUT instruments like the Butterfly iQ offer mobile device integration, with the ultrasound technology entirely housed within the handle and unattached to a display unit. These improvements have expanded the availability of ultrasonic devices and could lead to greater utility in many medical settings such as resource-limited communities [15]. Table 1 compares handheld and portable ultrasonic devices, including Butterfly iQ, though not all technologies listed are approved for ophthalmic use. This study systematically compares Butterfly iQ scans with scans from conventional ophthalmic ultrasound (COU) units: the 10MHz EyeCubed v3 (Ellex Inc., Adelaide, AUS) and 10MHz Accutome B-Scan Pro (Keeler, Malvern, PA). Fig 1 illustrates example images obtained from a healthy volunteer using the 10MHz EyeCubed v3 and Butterfly iQ devices.

## Methods

To qualitatively compare the axial and lateral resolution of Butterfly iQ and COU, phantom models of high (polymer molds in gelatin) and low (gelatin imprints filled with water) acoustic reflectivity were designed and imaged. Designs of the polymer molds (Molds A, B, C, and D) and diagrams of the phantoms produced using the molds are demonstrated in Fig 2. Mold A was designed to assess lateral resolution with decreasing lateral feature separation. Mold B offers features with decreasing width to evaluate lateral resolution. To examine axial

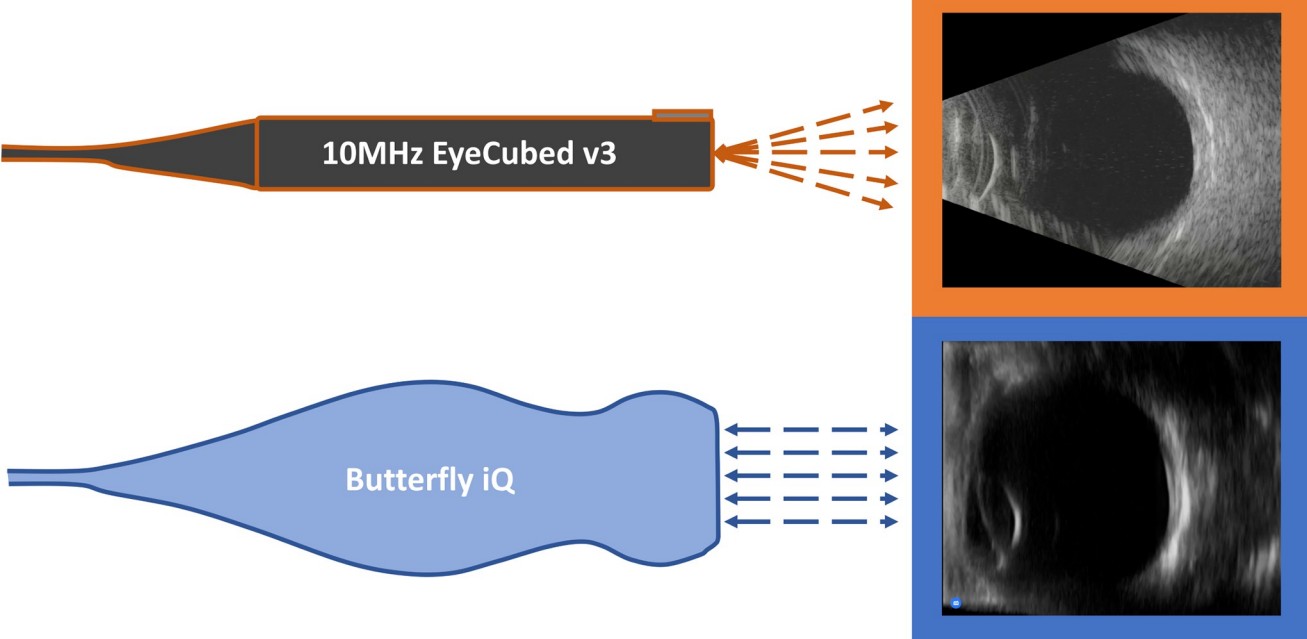

**Fig 1. Diagrams of the 10MHz EyeCubed v3 and Butterfly iQ devices and example images.** Images obtained from a healthy volunteer with the 10MHz EyeCubed v3 and Butterfly iQ devices are outlined in orange and blue respectively.

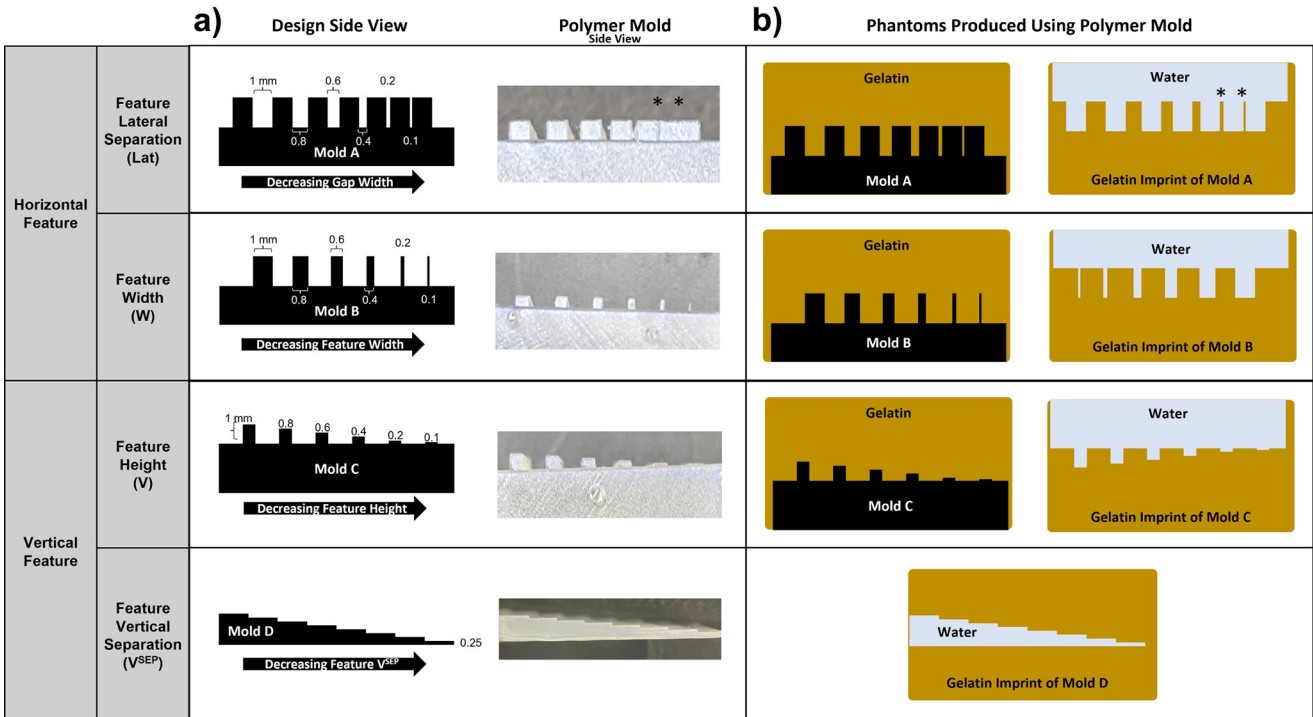

**Fig 2. Designs of the polymer molds and diagrams of the phantoms produced. a)** Polymer mold designs and side profile images of the printed polymers. Mold A was designed to compare lateral resolution with decreasing feature lateral separation. Mold B also compared lateral resolution but with decreasing feature width. Mold C examined axial resolution with decreasing feature heights. Mold D also investigated axial resolution by decreasing vertical separation of features. **b)** Diagrams of phantoms produced using the polymer molds. The molds were embedded in gelatin and imaged as high acoustic reflectivity phantoms. The molds were subsequently removed from the gelatin, leaving a gelatin imprint of the mold. Gelatin imprints from each mold were submerged in water and imaged as low acoustic reflectivity phantoms. *Correspond to features that were not preserved during removal of the polymer mold and were therefore unavailable for comparison.

resolution, Mold C was created with decreasing feature heights. Finally, Mold D investigated the axial resolution by containing features with decreasing vertical separation. Polymer molds were printed with ultraviolet-cured resin (Form 3B, FormLabs, Somervile, MA). Polymer molds were submerged in Knox Gelatin (Treehouse Foods, Oak Brook, IL) that was mixed with a gelatin-water ratio of 5:1 by weight to approximate the acoustic properties of vitreous humor (Fig 2B) [16]. Gelatin imprints were created by removing encased polymer molds, leaving a negative imprint of the surface topography (Fig 2B). The gelatin imprints were subsequently submerged in water for imaging. To compare axial resolution between Butterfly iQ and COU when passing through multiple layers of attenuating media, additional phantoms were created containing three paper sheets attached with double-sided tape along the perimeter to compose a paper stack (S1 Fig) with separation between paper sheets. These paper stacks were submerged in Knox Gelatin, as described above, and imaged for qualitative comparison. All gelatin phantoms were refrigerated at 5°C for at least 10 hours.

The Butterfly iQ probe was connected to an iPad Pro (Apple Inc., Cupertino, CA) for live imaging and data collection. The Butterfly iQ ophthalmic imaging preset (Butterfly iQ-Oph) was used and compared against the 10MHz Eye Cubed v3 ophthalmic ultrasonography unit to image polymer molds and gelatin imprints. Ultrasound images of the phantoms were independently reviewed by the authors, comprised of a team including a retina specialist, two postdoctoral researchers, and three medical students. Phantom images were examined for identification of features designed to test for axial and lateral resolution.

For a clinical imaging comparison, Institutional Review Board (HS#2019–5254) approval was obtained from the University of California, Irvine, and the study was conducted in accordance with the Declaration of Helsinki. Seven study participants provided written informed consent, and the study was conducted in a HIPAA-compliant manner. The recruitment period for this study was from 4/1/2020 to 12/31/2020. Only patients with a clinical indication were imaged with both the Butterfly iQ and the Accutome 10MHz B-Scan Pro. Four imaging presets (Musculoskeletal (MSK), Musculoskeletal-Soft Tissue (MSK-ST), Nerve (N), and Pediatric Lung (PL)) were selected from the Butterfly iQ software library. The Butterfly iQ-Oph imaging preset was not available in the United States when conducting this clinical imaging experiment. Therefore, settings were optimized for other organ presets during clinical evaluation. During imaging, one minute video clips were taken using both ultrasound probes. Still images were then exported, de-identified, and cropped.

Randomized images from study participants were presented in a questionnaire to three different ophthalmologists at the University of California, Irvine, who were blinded to whether the images were captured with the Butterfly iQ or the conventional piezoelectric ultrasound Accutome-B Scan Pro. Physician graders rated images on a ten-point Likert scale according to resolution, detail, image quality, and diagnostic confidence. A score of 10 represents an image demonstrating the highest quality of the criterion being measured, whereas a score of 1 represents the lowest quality. Resolution was defined as "sharpness of the image and lack of haziness"; detail as "clarity of outlines, how well structures and boundaries are defined"; image quality as "overall image assessment (e.g., absence of noise, contrast between structures)"; and diagnostic confidence as "confidence in making clinical decisions based on image." Pathologic conditions presented included repaired retinal detachment with silicon oil tamponade, vitreous hemorrhage, and tractional retinal detachment. Kruskal-Wallis analysis with Mann-Whitney U pairwise tests were utilized to compare survey responses between Butterfly iQ and COU. All statistical analyses were performed with SPSS Statistics 27 (IBM, Armonk, NY). P-values less than 0.05 were considered significant.

## Results

Ultrasound images of polymer molds within gelatin, representing high acoustic reflectivity phantoms, and the corresponding gelatin imprints submerged in water after mold removal, representing low acoustic reflectivity phantoms are shown in Fig 3. The authors unanimously agreed on which phantom features were identified on imaging. Except for Mold A, both Butterfly iQ and COU successfully identified the smallest available phantom features, down to 0.1 mm, for all phantoms tested (Fig 3). Mold A contained features separated by lateral gaps of decreasing size, ranging from 1 mm to 0.1 mm (Fig 2). The 0.2 and 0.1 mm gaps in Mold A were not visualized by either ultrasound modality, with the smallest resolved gap being 0.4 mm (Fig 3A). Differences in visualizing Mold A's 0.4 mm gap between Butterfly iQ and COU are highlighted by the red arrows in Fig 3A. While COU was able to resolve the openings of the 0.2 and 0.1 mm gaps of Mold A, the features were incompletely visualized. In addition, when Mold A was removed from its corresponding gelatin, the smallest features were not preserved. The red arrowheads in Fig 3D indicate the features that were damaged during mold removal and were unavailable for comparison.

For comparison of axial resolution when passing through multiple layers of attenuating media, phantoms created with paper stacks (S1 Fig) were imaged with both Butterfly iQ and COU. Qualitatively, Butterfly iQ and COU both identified all three layers of paper with similar image qualities, indicating comparable axial resolution performance in this setting.

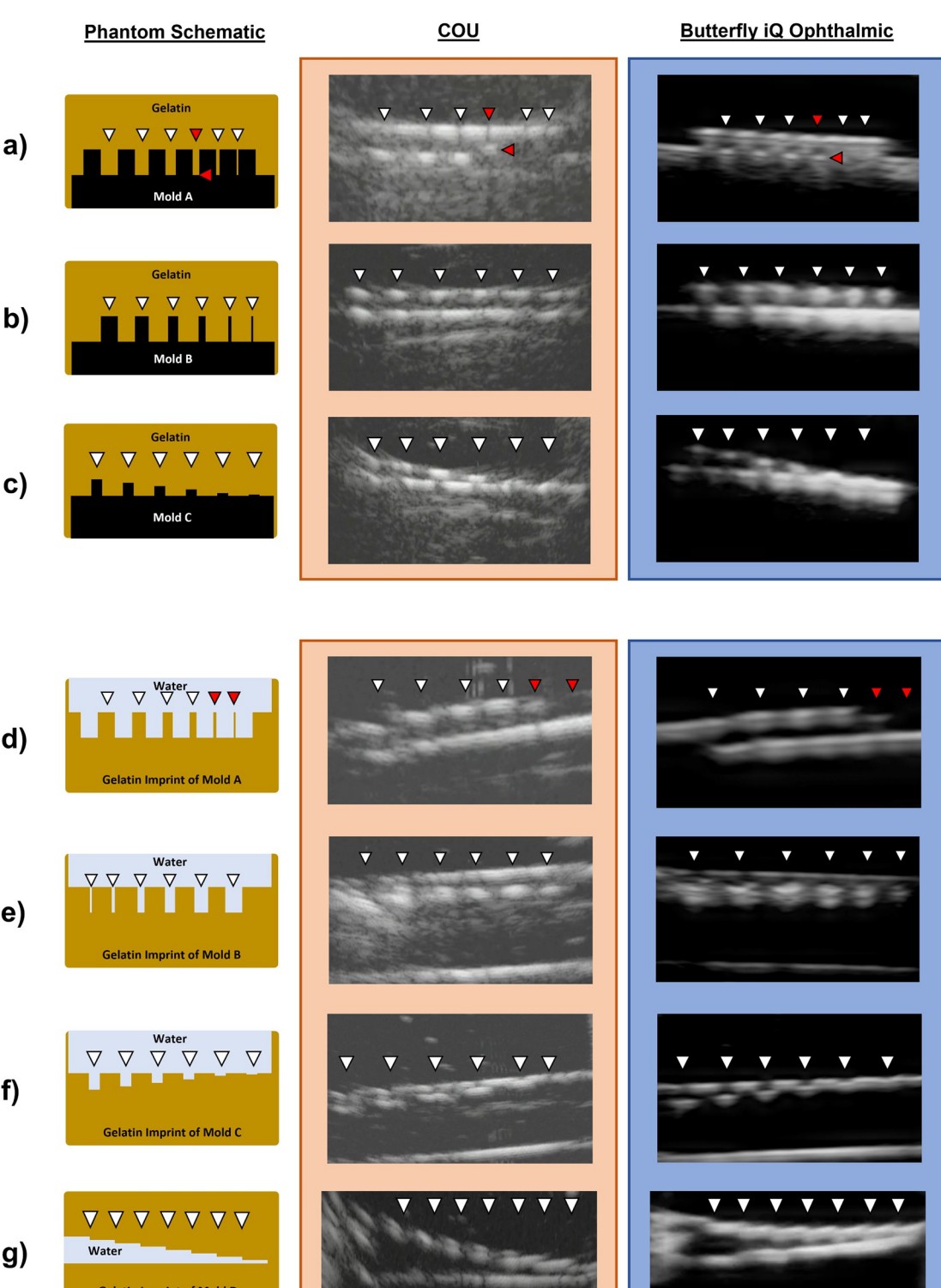

**Fig 3. Diagrams of phantoms and images taken with Butterfly iQ and COU.** High acoustic reflectivity polymer molds (designs listed in Fig 1) were embedded in gelatin and compared in rows a-c. Low acoustic reflectivity gelatin imprints, created by removing the molds, are compared in rows d-g. White arrowheads indicate the horizontal positions of phantom features on the diagrams and corresponding ultrasound images. The red arrowheads in row a highlight the differences observed in the 0.4 mm lateral feature separation in Mold A between Butterfly iQ (blue) and COU (orange). The red arrowheads in row d indicate gelatin features that were not preserved upon removing Mold A from its embedded gelatin.

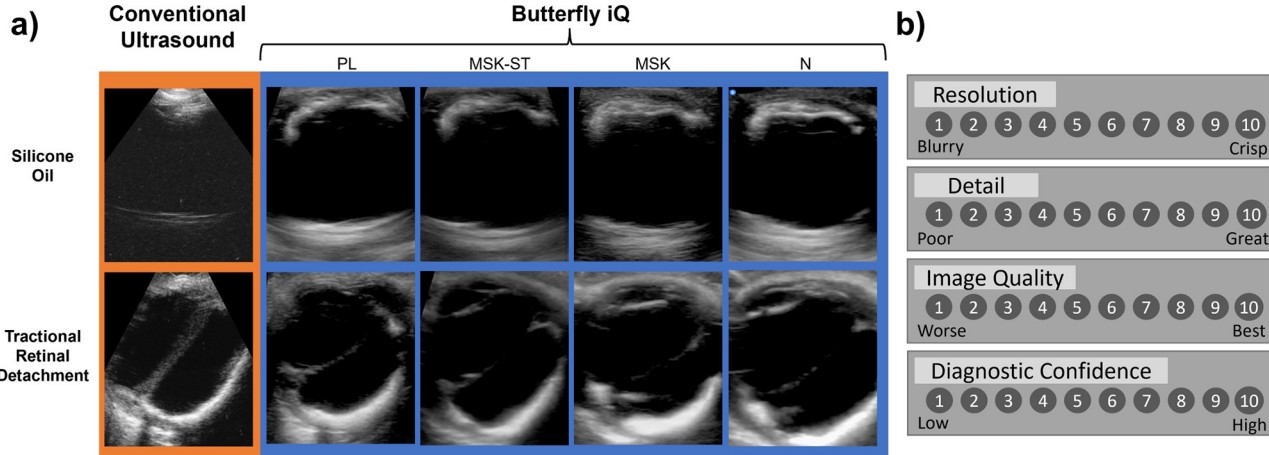

**Fig 4. Example images and Likert scale featured in the survey. a)** Select images of pathologies featured in the survey, including intraocular silicone oil and tractional retinal detachment. COU images are seen in the orange outline, whereas Butterfly iQ images with the PL, MSK-ST, MSK, and N presets are displayed with the blue outline. **b)** Likert scale used by graders to evaluate images for resolution, detail, image quality, and diagnostic confidence.

To evaluate clinical performance of both ultrasound devices, three blinded retina specialists used a Likert scale (Fig 4B) to compare image characteristics of study participants (Fig 4A). Two graders reported no statistically significant differences between Butterfly iQ and COU for all image qualities surveyed (Table 2). Grader 2 showed consistent scoring of COU images higher than any Butterfly iQ preset regarding resolution, detail, and diagnostic confidence.

### Values with statistically significant difference are in bold

Grader 2 maintained a statistically significant preference for COU's resolution, detail, and diagnostic confidence, while not favoring either modality for image quality. Grader 1 and Grader 3 showed no statistically significant preference for Butterfly iQ or COU across all qualities surveyed.

## Discussion

In this study, Butterfly iQ was compared with conventional piezoelectric ophthalmic ultrasonography. In our comparison, polymer and gelatin phantoms highlighted comparable imaging capabilities of both Butterfly iQ and COU in imaging the depth and width of small features (0.1 mm) of high and low acoustic reflectivity. However, further experimentation would be required to reliably form and image smaller tissues (<0.1 mm). Furthermore, the analysis of the survey demonstrated one of three retina specialists showing a higher preference for COU

**Table 2. Results of Kruskal-Wallis analysis on survey responses.**

|  | **Grader 1** | **Grader 2** | **Grader 3** |
|---|---|---|---|
| Resolution | 0.587 (2.826) | **0.047 (9.647)** | 0.887 (1.147) |
| Detail | 0.533 (3.148) | **0.022 (11.421)** | 0.897 (1.085) |
| Image Quality | 0.623 (2.619) | 0.608 (2.706) | 0.828 (1.493) |
| Diagnostic Confidence | 0.726 (2.051) | **0.045 (9.723)** | 0.968 (0.558) |

p-value (H statistic)
Four degrees of freedom

related to resolution, detail, and diagnostic confidence (Table 2). Remaining graders showed no difference amongst all modalities and imaging settings portrayed, demonstrating similar imaging quality between Butterfly iQ and COU. One grader consistently favoring COU may be a consequence of recognition bias as the graders could independently identify which ultrasound instrument produced each image based on image characteristics (note differences visible in Fig 4). One highlighted advantage of using Butterfly iQ is the ability to acquire an image in low density fluids like silicone oil, whereas the image quality in COU is well known to be dramatically diminished in the presence of intraocular silicone oil (Fig 4).

Our comparison of Butterfly iQ, a portable CMUT ultrasound technology, and conventional piezoelectric ultrasound used in ophthalmology revealed minor differences in phantom imaging and survey results. Early inquiries comparing the use of piezoelectric probes with research-grade CMUTs for medical imaging have shown similar imaging resolution capabilities [17]. Currently, Butterfly iQ is the only commercially available CMUT probe for clinical use since the L38-22v and L22-8v (Kolo Medical, San Jose, CA) probes (Table 1) require an interfacing system (Vantage System, Verasonics, Kirkland, WA), which does not have FDA clearance as a medical device.

Additionally, recent comparisons of Butterfly iQ to piezoelectric transducers have not compared modalities for ophthalmic use. Sabbadini et al. compared Butterfly iQ to Logiq S7 Expert (GE Healthcare, Chicago, IL) to assess the total time to complete the Ultrasound Hypotension Protocol (UHP) and conferred with licensed sonographers to determine that modalities were equivalent for diagnostic purposes [18]. Similarly, Dewar et al. assessed Butterfly iQ and Sparq (Phillips, Amsterdam, NL) equivalence in cardiac imaging for Rapid Ultrasound for Hypotension and Shock (RUSH) imaging. Though this study used three physician graders, the scoring sheet utilized in their study was also a binary answer choice (yes/no) for "adequate quality for interpretation" [19]. Instead of using a binary answer choice, our survey utilized a 10-point Likert scale to delineate fine score differences between physician graders.

Bennett et al. demonstrated insignificant differences between Butterfly iQ and Venue GO (GE Healthcare) point-of-care-ultrasound for lung aeration scoring (0–36) in infections due to Severe Acute Respiratory Syndrome Corona Virus 2 (SARS-CoV-2) [20]. This grading scale was a numerical score; however, it was a qualitative assessment of imaging capability as higher point values were assigned and added for worsening appearance of aeration in specific areas of the lung [20]. Additionally, the authors tested for agreement of scores derived from both ultrasound modalities, instead of specific criteria of images. In our study, the phantom models contained features with known dimensions that allowed for resolution comparison between both ultrasound technologies.

One of the limitations of this study is that we assumed comparable performance between two different COU instruments. We used the Ellex EyeCubed v3 on phantom models to characterize resolution and the Accutome B-scan Pro to acquire the clinical images included in the qualitative evaluation by physician graders. Further, we evaluated a small clinical sample and definitive conclusion of equivalence or non-inferiority between Butterfly iQ and COU would require a larger pool of disease states and expanded pool of blinded graders to quantify sensitivity and specificity when detecting pathology.

## Conclusions

Our comparison of Butterfly iQ and COU demonstrated similar resolution in phantom imaging, with both modalities achieving qualitative imaging resolution down to 0.1 mm. In the context of clinical imaging, two out of three blinded retina specialists showed no statistically significant preference for COU over Butterfly iQ when presented with images of ophthalmic

pathologies. These results underscore the significance of Butterfly iQ as a unique handheld alternative to COU. Butterfly iQ's affordability and portability may lower the barrier to entry for ophthalmic ultrasound imaging and provide more comprehensive ophthalmic care in resource-limited communities. The findings in this study justify expanded clinical studies comparing Butterfly iQ and other CMUT devices with COU.

## Supporting information

**S1 Fig. Diagram and ultrasound images of paper phantom model. a)** Diagram of model with three papers (black) each separated by three layers of tape (white) and gelatin (yellow). **b)** COU imaging is outlined in orange and Butterfly iQ imaging is outlined in blue.
(TIF)

**S1 Data. Polymer phantom CAD file.** CAD file for the 3D printed structure used to generate Molds A, B, and C (Figs 2 and 3).
(STL)

**S2 Data. Polymer phantom CAD file.** CAD file for the 3D printed structure used to generate Mold D (Figs 2 and 3).
(STL)

**S1 Dataset. Clinical survey results and statistical analysis.**
(XLSX)

**S1 File. Questionnaire used to survey three retina specialists comparing clinical images from Butterfly iQ and COU.**
(PDF)

## Author Contributions

**Conceptualization:** Jack O. Thomas, Josiah K. To, Anderson N. Vu, Andrew W. Browne.

**Data curation:** Jack O. Thomas, Josiah K. To, Anderson N. Vu, David Horton, Ermin Dzihic.

**Formal analysis:** Josiah K. To, Anderson N. Vu, Andrew W. Browne.

**Funding acquisition:** Andrew W. Browne.

**Investigation:** Jack O. Thomas, Josiah K. To, Anderson N. Vu, David Horton, Ermin Dzihic, Andrew W. Browne.

**Methodology:** Jack O. Thomas, Josiah K. To, Anderson N. Vu, David Horton, Ermin Dzihic, Andrew W. Browne.

**Project administration:** Andrew W. Browne.

**Resources:** Andrew W. Browne.

**Software:** Andrew W. Browne.

**Supervision:** Andrew W. Browne.

**Validation:** Jack O. Thomas, Josiah K. To, Anderson N. Vu, David Horton, Ermin Dzihic, Andrew W. Browne.

**Visualization:** Jack O. Thomas, Josiah K. To, Anderson N. Vu, David Horton, Ermin Dzihic, Andrew W. Browne.

**Writing – original draft:** Jack O. Thomas, Josiah K. To, Anderson N. Vu, Andrew W. Browne.

**Writing – review & editing:** Jack O. Thomas, Josiah K. To, Anderson N. Vu, David Horton, Ermin Dzihic, Andrew W. Browne.

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
