## [Decision Letter · Decision Letter 0]

3 Oct 2023

PONE-D-23-27765Imaging performance of portable and conventional ultrasound imaging technologies for ophthalmic applicationsPLOS ONE

Dear Dr. Thomas,

Thank you for submitting your manuscript to PLOS ONE. After careful consideration, we feel that it has merit but does not fully meet PLOS ONE’s publication criteria as it currently stands. Therefore, we invite you to submit a revised version of the manuscript that addresses the points raised during the review process.

We look forward to receiving your revised manuscript.

Kind regards,

Ireneusz Grulkowski, PhD

Academic Editor

PLOS ONE

Journal Requirements:

Whilst you may use any professional scientific editing service of your choice, PLOS has partnered with both American Journal Experts (AJE) and Editage to provide discounted services to PLOS authors. Both organizations have experience helping authors meet PLOS guidelines and can provide language editing, translation, manuscript formatting, and figure formatting to ensure your manuscript meets our submission guidelines. To take advantage of our partnership with AJE, visit the AJE website (http://aje.com/go/plos) for a 15% discount off AJE services. To take advantage of our partnership with Editage, visit the Editage website (www.editage.com) and enter referral code PLOSEDIT for a 15% discount off Editage services. If the PLOS editorial team finds any language issues in text that either AJE or Editage has edited, the service provider will re-edit the text for free.

A clean copy of the edited manuscript (uploaded as the new *manuscript* file).

Reviewers' comments:

Reviewer's Responses to Questions

**Comments to the Author**

1. Is the manuscript technically sound, and do the data support the conclusions?

Reviewer #1: Partly

2. Has the statistical analysis been performed appropriately and rigorously? 

Reviewer #1: Yes

3. Have the authors made all data underlying the findings in their manuscript fully available?

Reviewer #1: Yes

4. Is the manuscript presented in an intelligible fashion and written in standard English?

Reviewer #1: Yes

5. Review Comments to the Author

Reviewer #1: I would like to congratulate the authors on completing this study and well-written manuscript. Overall, this is an interesting pilot or proof-of-concept study evaluating the Butterfly iQ compared to a conventional ultrasound device, specifically for ocular applications.

Some specific comments:

I suggest changing the terminology to focus more specifically on the Butterfly iQ itself rather than CMUT in general, since that is the only one tested, and the only one currently approved for ocular use.

Line 44 – take out the second “technology”, perhaps “increased portability with decreased equipment cost”

Reconsider the use ofthe abbreviation BiQ. Have never seen that used.

Move the sentence starting on line 63 “These… 2D gray-scale image” to the second sentence of the same paragraph because all ultrasound waves, regardless of generating technique, function this way. Then contrast how the waves are produced.

Materials and Methods

The authors went into great detail describing the making of the phantoms, but then didn’t really describe how they were to be systematically evaluated. Then they are discussed again in the results, but it is still unclear who did the evaluating, or by what standards.

The results paragraph regarding the phantoms and corresponding Figure 3 are difficult to interpret.

I would also consider re-writing the conclusion section. This is a very generic conclusion that could have been made without the study. I would include statements addressing the similar imaging resolution on the custom phantoms, and the statistical agreement between 2/3 blinded reviewers

6. PLOS authors have the option to publish the peer review history of their article (what does this mean?). If published, this will include your full peer review and any attached files.

Reviewer #1: No

---

## [Author Response · Author response to Decision Letter 0]

30 Nov 2023

We thank the reviewers for their efforts in helping us improve our manuscript. A detailed, point-by-point reply to each of the comments made by the reviewers is enclosed in our attached file "Response to Reviewers".

---

## [Decision Letter · Decision Letter 1]

12 Feb 2024

PONE-D-23-27765R1Imaging performance of portable and conventional ultrasound imaging technologies for ophthalmic applicationsPLOS ONE

Dear Dr. Thomas,

Thank you for submitting your manuscript to PLOS ONE. After careful consideration, we feel that it has merit but does not fully meet PLOS ONE’s publication criteria as it currently stands. Therefore, we invite you to submit a revised version of the manuscript that addresses the points raised during the review process.

As you can see from the enclosed reviews, the reviewers find your manuscript potentially suitable for publication in PLoS One. However, one of them raised a specific issue that must be addressed before the final acceptance of your manuscript. Therefore, I am requesting that you submit a revised version of this manuscript to address the comments. To help me expedite processing, please explicitly address the questions raised by the reviewer in your cover letter and also indicate the changes made in the manuscript.

We look forward to receiving your revised manuscript.

Kind regards,

Bing Xu, PhD

Academic Editor

PLOS ONE

Journal Requirements:

Reviewers' comments:

Reviewer's Responses to Questions

**Comments to the Author**

1. If the authors have adequately addressed your comments raised in a previous round of review and you feel that this manuscript is now acceptable for publication, you may indicate that here to bypass the “Comments to the Author” section, enter your conflict of interest statement in the “Confidential to Editor” section, and submit your "Accept" recommendation.

Reviewer #1: All comments have been addressed

Reviewer #2: (No Response)

2. Is the manuscript technically sound, and do the data support the conclusions?

Reviewer #1: Yes

Reviewer #2: Partly

3. Has the statistical analysis been performed appropriately and rigorously? 

Reviewer #1: Yes

Reviewer #2: No

4. Have the authors made all data underlying the findings in their manuscript fully available?

Reviewer #1: Yes

Reviewer #2: Yes

5. Is the manuscript presented in an intelligible fashion and written in standard English?

Reviewer #1: Yes

Reviewer #2: Yes

6. Review Comments to the Author

Reviewer #1: The authors have successfully implemented the recommendations and improvements suggested in the first round of edits. The paper is improved in clarity and presentation.

Reviewer #2: The manuscript by To et al. assessed the imaging performance of the Butterfly iQ (BiQ), a portable ultrasound probe, in comparison to conventional ophthalmic ultrasound (COU) for ophthalmic imaging. Custom phantom molds were created by the researchers, and both BiQ and COU were used to capture images for a spatial resolution comparison. Additionally, a survey involving three retina specialists was conducted to evaluate images of pathological conditions from human subjects obtained with both probes. The findings indicated that BiQ and COU exhibited similar imaging capabilities for small axial and lateral phantom features. While one specialist favored COU, the overall preference did not significantly lean towards either probe. This study suggests that portable ultrasound probes, like BiQ, present a cost-effective alternative to COU, offering comparable imaging resolution and indicating their potential usefulness in routine eye care. I support the acceptance of this manuscript after the authors address the following issues.

1. The primary issue lies in the study's small sample size and limited clinical assessment. With only seven participants and three retina specialists included, the findings may not be representative to a wider population. It is suggested that the authors provide rationale for their choice of sample size.

2. As for the clinal evaluation, it would be beneficial if the authors incorporate analysis using image processing software in addition to the specialists' ratings.

3. The authors should provide the full names for the abbreviations, such as PL, MSK-ST, MSK, and N.

7. PLOS authors have the option to publish the peer review history of their article (what does this mean?). If published, this will include your full peer review and any attached files.

Reviewer #1: No

Reviewer #2: No

---

## [Author Response · Author response to Decision Letter 1]

20 Feb 2024

We have uploaded a detailed, point-by-point response to the reviewers in our letter "Response to Reviewers".

---

## [Editor Report · Decision Letter 2]

28 Feb 2024

Imaging performance of portable and conventional ultrasound imaging technologies for ophthalmic applications

PONE-D-23-27765R2

Dear Dr. Thomas,

We’re pleased to inform you that your manuscript has been judged scientifically suitable for publication and will be formally accepted for publication once it meets all outstanding technical requirements.

Kind regards,

Bing Xu, PhD

Academic Editor

PLOS ONE
---

## [Editor Report · Acceptance letter]

30 Apr 2024

PONE-D-23-27765R2 

PLOS ONE

Dear Dr. Thomas, 

I'm pleased to inform you that your manuscript has been deemed suitable for publication in PLOS ONE. Congratulations! Your manuscript is now being handed over to our production team.

Kind regards, 

on behalf of

Dr. Bing Xu 

Academic Editor

PLOS ONE